# Degenerate sequence-based CRISPR diagnostic for Crimean–Congo hemorrhagic fever virus

Hongzhao Li[1], Alexander Bello[2], Greg Smith[1], Dominic M. S. Kielich[3], James E. Strong[2,3,4], Bradley S. Pickering[1,3,5] *

1 National Centre for Foreign Animal Disease, Canadian Food Inspection Agency, Winnipeg, Canada, 2 National Microbiology Laboratory, Public Health Agency of Canada, Winnipeg, Canada, 3 Department of Medical Microbiology and Infectious Diseases, College of Medicine, Faculty of Health Sciences, University of Manitoba, Winnipeg, Canada, 4 Department of Pediatrics & Child Health, College of Medicine, Faculty of Health Sciences, University of Manitoba, Winnipeg, Canada, 5 Iowa State University, College of Veterinary Medicine, Department of Veterinary Microbiology and Preventive Medicine, Ames, Iowa, United States of America

* bradley.pickering@inspection.gc.ca

**Data Availability Statement:** All relevant data are within the manuscript and its Supporting Information files.

## Abstract

CRISPR (clustered regularly interspaced short palindromic repeats), an ancient defense mechanism used by prokaryotes to cleave nucleic acids from invading viruses and plasmids, is currently being harnessed by researchers worldwide to develop new point-of-need diagnostics. In CRISPR diagnostics, a CRISPR RNA (crRNA) containing a "spacer" sequence that specifically complements with the target nucleic acid sequence guides the activation of a CRISPR effector protein (Cas13a, Cas12a or Cas12b), leading to collateral cleavage of RNA or DNA reporters and enormous signal amplification. CRISPR function can be disrupted by some types of sequence mismatches between the spacer and target, according to previous studies. This poses a potential challenge in the detection of variable targets such as RNA viruses with a high degree of sequence diversity, since mismatches can result from target variations. To cover viral diversity, we propose in this study that during crRNA synthesis mixed nucleotide types (degenerate sequences) can be introduced into the spacer sequence positions corresponding to viral sequence variations. We test this crRNA design strategy in the context of the Cas13a-based SHERLOCK (specific high-sensitivity enzymatic reporter unlocking) technology for detection of Crimean–Congo hemorrhagic fever virus (CCHFV), a biosafety level 4 pathogen with wide geographic distribution and broad sequence variability. The degenerate-sequence CRISPR diagnostic proves functional, sensitive, specific and rapid. It detects within 30–40 minutes 1 copy/μl of viral RNA from CCHFV strains representing all clades, and from more recently identified strains with new mutations in the CRISPR target region. Also importantly, it shows no cross-reactivity with a variety of CCHFV-related viruses. This proof-of-concept study demonstrates that the degenerate sequence-based CRISPR diagnostic is a promising tool of choice for effective detection of highly variable viral pathogens.

**Funding:** This work was supported by funding to BSP, provided from the Canadian Safety and Security Program (Grant number: CSSP-2018-CP-2341). The funder's URL is: https://science.gc.ca/eic/site/063.nsf/eng/h_5B5BE154.html. The funder had no role in study design, data collection and analysis, decision to publish, or preparation of the manuscript.

**Competing interests:** The authors have declared that no competing interests exist.

## Author summary

Several types of CRISPR-based molecular diagnostics, currently under extensive development, hold great promise for field-deployable diagnosis of infectious diseases. These methods share a core mechanism of target recognition, where a CRISPR RNA (crRNA) contains a pathogen-specific "spacer" sequence that binds to the counterpart on the genetic material of the targeted pathogen. The recognition mechanism can sometimes be interfered with by sequence mismatches between the spacer and target, which could arise from variations in the target sequence. One of the variable targets of concern is Crimean–Congo hemorrhagic fever virus (CCHFV), which demonstrates broad sequence diversity and evolves into several genetic clades. To address the viral sequence diversity, we have developed a "degenerate" sequence-based CRISPR strategy. It introduces mixed nucleotide types at sequence positions of a crRNA spacer corresponding to variations in the target pathogen. The resulting assay detects all CCHFV clades effectively. These findings indicate that degenerate-sequence CRISPR is a feasible way to address highly variable diagnostic targets.

## Introduction

Rapid reliable pathogen detection methods are critical for effective public health control measures, in order to prevent the further spread of infectious diseases and provide early diagnosis and treatment of patients [1]. Quantitative polymerase chain reaction (qPCR) is currently the gold standard diagnostic test for early-stage infection based on highly sensitive and specific nucleic acid detection [2,3]. However, qPCR is resource-demanding and requires high temperature thermal cycling, expensive and bulky instruments, sophisticated technical procedures and professionally trained operators. Its limited availability, largely from centralized laboratories, together with a slow turnaround, does not meet the need for wide-range and rapid testing during a fast spreading outbreak [2,3].

A number of isothermal nucleic acid detection methods are emerging as potential supplements or alternatives to qPCR and are candidate tools for on-site, point-of-need (PON) testing [1]. PON diagnostics are anticipated to be broadly available, rapid, portable, easy to perform (by a non-professional with minimal or no training), and preferably sensitive and specific, at levels similar to or even higher than qPCR. They can be deployed in extensive scenarios, for example, in medical laboratories to supplement testing at overwhelmingly high demand, or directly in the public and communities for mass testing (in airports, doctor's offices, pharmacies, schools, or even homes, or by any individuals themselves). Further, they are particularly suitable for mobile testing (in mobile labs set up in the field, or on the move even during sample collection and transportation processes), frequent repeated testing, or deployment in resource-constrained countries and regions, including rural and remote areas where access to diagnostic testing is unavailable [1]. Among isothermal rapid molecular detection techniques, recombinase polymerase amplification (RPA) and loop mediated isothermal amplification (LAMP) platforms have so far been the best studied and most popularly used in the development of PON diagnostics [1,3,4]. These can be optimized to reach high sensitivity, although the sensitivity is typically lower than that of qPCR, and can sometimes suffer from nonspecific amplification leading to false positive results, especially when non-sequence-specific probes are used [3,5]. It is expected that improvement can be made through their coupling to an additional, sensitive and specific detection method [5].

CRISPR (clustered regularly interspaced short palindromic repeats), is a prokaryotic adaptive immune system against viral and plasmid infections. A recent explosion of research efforts to adapt the system to new, isothermal nucleic acid detection tools for infectious diseases has begun, particularly spurred by the COVID-19 pandemic [5–18]. In a CRISPR diagnostic reaction, a CRISPR RNA (crRNA) recognizes a target nucleic acid sequence and bridges the activation of a CRISPR-associated (Cas) protein effector, Cas9, Cas12a, Cas12b, Cas13a or Cas14 [19–23]. The most promising and prevalent of these, Cas13a, Cas12a and Cas12b, upon activation not only degrade their specific targets but also exhibit a collateral and indiscriminate RNase (Cas13a) or DNase (Cas12a and Cas12b) activity that trans-cleaves bystander RNA or DNA reporter molecules, leading to enormous amplification of fluorescent or colorimetric signals. A variety of new diagnostic platforms are emerging, both highly sensitive and specific, based on incorporation of CRISPR with other isothermal nucleic acid amplification methods, including RPA and LAMP. They hold great promise for further optimization and formatting into the ideal PON diagnostic products as anticipated above [5–18]. A representative system of these, also known as SHERLOCK (specific high-sensitivity enzymatic reporter unlocking), integrates the three major known lower-temperature (37˚C or room/ambient temperature) nucleic acid amplification techniques: RPA, T7 transcription and CRISPR/Cas13a [19]. Apart from having all the other advantages of a desired PON diagnostic, SHERLOCK can be operated with body heat or left-alone in an environment with a similar temperature. This unique feature makes it the best fit for diagnostic scenarios without access to (ample) electricity, such as in rural or remote regions and on-the-go testing.

CRISPR target recognition, through basepairing between a "spacer" sequence on the crRNA and its complementary counterpart on the target RNA or DNA, can sometimes be disrupted by sequence mismatches between the spacer and target [19,24–27], whereas the effect of different types of mismatches and the underlying mechanisms remain to be understood. It is thus an open possibility that some types of mutations in the targeted genomic region of a pathogen could allow escape from detection. This is especially problematic in RNA viruses, which have evolved to possess enormous amounts of sequence diversity, thanks to poor proofreading during the replication of their genome, apart from other mechanisms [28]. Among these are emerging or re-emerging high consequence pathogens such as Crimean-Congo hemorrhagic fever virus (CCHFV) [29], human immunodeficiency virus (HIV) [30] and highly pathogenic influenza virus [31]. Even in SARS-CoV-2, which has newly entered the human population, sequence diversification has already raised major concerns [32–37].

Traditional CRISPR diagnostics, which use crRNAs with a single constant spacer sequence, can be aimed at a conserved region of the viral genome with relatively low extent of variations. The success to cover all (or most) viral variants can be achievable under the condition that a target region is available and identified where all (or most) versions of sequence variations are tolerated by the constant spacer sequence (as experimentally tested, on a case-by-case basis). Alternatively, we are seeking a general way that readily covers all variants by design. In this study, we hypothesized that all viral variants are covered with certainty if mixed (degenerate) nucleotide types are incorporated into the crRNA spacer, matching all the variation types found in the target viral sequences. We tested this new CRISPR design strategy in the context of Cas13a-based SHERLOCK technology for detection of CCHFV.

CCHFV is a biosafety level 4 (BSL4) pathogen, and has been listed by WHO as one of the eight top priority pathogens expected to cause severe outbreaks in the near future [38]. Infection by the virus can result in a severe hemorrhagic fever disease in humans, with a fatality rate ranging from 5% to 30%, and up to above 60%, depending on reports of cases from different countries/regions [29]. CCHFV is the most geographically widespread pathogen among tick-borne viruses and the second most geographically widespread pathogen among arboviruses

(after Dengue virus). Its distribution involves Asia (from Western China to the Middle East and Turkey), Europe (Eastern and Southeastern countries) and Africa (majority of the continent). The vast geographic range reflects that CCHFV hosts tolerate broadly diverse environments [29]. The virus is also considered an emerging pathogen, with cases being reported in new areas such as west Europe. It is an open possibility that CCHFV may expand far beyond traditional endemic regions, potentially contributed to by emergence of favorable ecological environment for viral hosts due to climate change, introduction of infected ticks by migratory birds or introduction of infected animals by livestock trade [29,39–42].

Belonging to the genus Nairovirus in the family Bunyaviridae, CCHFV is an enveloped virus with a negative-sense RNA genome consisting of the small (S), medium (M) and large (L) segments. In these segments sequence divergence among viral strains was found to be 20, 31 and 22%, respectively, representing the greatest degree of sequence diversity of any arbovirus. The S segment (or the S gene), which encodes the viral nucleoprotein and is thus also called the N gene, is the most conserved and the most often targeted by diagnostic development among CCHFV genomic segments. Based on the S segment sequences, phylogenetic trees demonstrated 6–7 genetic clades/lineages [29]. Using CCHFV as an example, this proof-of-principle study indicated that a degenerate sequence-based CRISPR diagnostic can successfully detect variable targets in a highly sensitive and specific manner.

## Materials and methods

### Biosafety statement

All experiments involving infectious CCHFV and Rift Valley fever virus were conducted in containment level 4 (CL4) and CL3+ laboratories, respectively, at the National Centre for Foreign Animal Diseases (NCFAD), Canadian Food Inspection Agency (CFIA), following the institutional standard operating procedures (SOPs).

### The CRISPR/Cas13a-based SHERLOCK assay for CCHFV detection

This was designed and performed by largely following a previously published two-step protocol [19]. For RNA virus detection, Step 1 is a combination of reverse transcription to convert RNA into DNA and recombinase polymerase amplification to amplify DNA (RT-RPA). Step 2 consists of T7 transcription to convert and amplify DNA into RNA followed by CRISPR/Cas13a reaction for RNA detection and signal amplification.

A master mix of the RT-RPA formulation included the following components: one TwistAmp pellet, 6.95 µl H2O, 29.5 µl TwistAmp rehydration buffer, 2.4 µl forward primer (10 uM), 2.4 µl reverse primer (10 uM), 1.25 µl RNase inhibitor (40 U/µl), 2.5 µl RevertAid reverse transcriptase (200 U/µl) and 2.5 µl TwistAmp magnesium acetate (280mM). The above master mix is divided into 9.5-µl aliquots, followed by addition of 0.5 µl template RNA to form a 10-µl reaction. The CRISPR/Cas13a detection reaction formulation (25 µl) included the following components: 15.4625 µl H2O, 0.5 µl HEPES (1M), 0.225 µl MgCl2 (1M), 1 µl rNTP mix (25 mM each), 0.625 µl RNase inhibitor (40 U/µl), 1.25 µl crRNA (100 ng/µl), 2.5 µl LwCas13a protein (63.3 ng/µl in storage buffer), 0.625 µl T7 RNA polymerase (50 U/µl), 1.5625 µl RNase Alert v2 (2 µM in 1× RNase Alert buffer) and 1.25 µl RT-RPA reaction. The name and commercial source of stock reagents used in these formulations (supplier, catalogue number) included: UltraPure DEPC-Treated Water (ThermoFisher Scientific, 750024), TwistAmp Basic RPA kit (TwistDx, TABAS03KIT), RNase inhibitor (New England BioLabs, M0314L), RevertAid reverse transcriptase (ThermoFisher Scientific, EP0442), HEPES (ThermoFisher Scientific, 15630080), MgCl$_2$ (ThermoFisher Scientific, AM9530G), rNTP mix (New England BioLabs, N0466L), NxGen T7 RNA Polymerase (Lucigen, 30223–1) and RNase Alert v2

(ThermoFisher Scientific, 4479768). RT-RPA primers, CRISPR RNAs (crRNAs) and synthetic CCHFV RNAs were custom-synthesized by Integrated DNA Technologies or GenScript. LwCas13a was in-house produced. The methods in designing RT-RPA primers, CRISPR RNAs (crRNAs) and synthetic CCHFV RNA and making LwCas13a are described below.

The RT-RPA reaction was first assembled on ice and then incubated at 42˚C on a GeneAmp PCR System 9700 (ThermoFisher Scientific) for 25 minutes. At 5 minutes post the start of incubation, the reaction was agitated by pipetting three times (in one to two seconds), and then allowed to complete the whole incubation time. The T7 amplification and CRISPR--Cas13a detection step was also first prepared on ice, and then incubated at 37˚C. CCHFV target signal was acquired every five minutes, using the FAM channel on a CFX96 Real-Time PCR System (Bio-Rad).

## Definition of copy number per μl (cp/μl) or copy number per reaction (cp/reaction) of the assay

Since the two-step CRISPR (SHERLOCK) assay consists of an RT-RPA pre-amplification reaction followed by a T7-Cas13a reaction, the definition of cp/μl or cp/reaction was based on the final reaction, the T7-Cas13a reaction:

CCHFV target RNA concentration (cp/μl) = Total copy number of target RNA that was input into the final T7-Cas13a reaction divided by volume (μl) of the final T7-Cas13a reaction.

Total copy number of CCHFV target RNA per reaction (cp/reaction) = Total copy number of target RNA that was input into the final T7-Cas13a reaction, which was 25 μl in volume.

## LwCas13a enzyme production and purification

Plasmid for LwCas13a protein expression was created by the Feng Zhang lab and deposited with Addgene (plasmid # 90097). LwCas13a was tagged with TwinStrep-SUMO and produced and purified as previously described with modifications [6]. Briefly, LwCas13a expression vector was transformed into Rosetta2 DE3 cells and incubated overnight at 37˚C on Luria broth (LB) agar plates with 100 μg/ml ampicillin. Starter culture was prepared by picking a transformant and incubating in 25 ml Terrific Broth (TB) with 100 μg/ml ampicillin overnight at 37˚C with shaking. Starter culture was used to inoculate 4L TB media with 100 μg/ml ampicillin and incubated with shaking at 37˚C until an optical density (OD, 600nm) of 0.4–0.6 was reached. The culture was then transferred to 4˚C for 30 minutes. LwCas13a protein expression was induced with 0.5mM IPTG and incubated in a refrigerated shaker incubator at 21˚C for 16 hours. The cells were pelleted and lysed with supplemented lysis buffer (20 mM Tris-HCl pH 8.0, 500 mM NaCl, 1 mM DTT, cOmplete ULTRA EDTA-free tablets (Sigma-Aldrich Canada Co., Oakville, ON, Canada), lysozyme, benzonase). Lysate was processed with sonication using amplitude of 100 for 1 second on and 2 seconds off, for a total of 10 minutes of sonication time on a VirTis Virsonic 600 Ultrasonic Cell Disruptor and lysate clarified by centrifugation for 1 hour at 10K RPM at 4˚C. Clarified supernatant was incubated with 5 ml Cytiva StrepTactin Sepharose High Performance Medium (Fisher Scientific) for 2 hours, added to an equilibrated 50 ml Bio-Rad Glass Econo-Column, and washed with unsupplemented lysis buffer (no cOmplete ULTRA EDTA-free tablets, no lysozyme, no benzonase). TwinStrep-SUMO tag was cleaved to release LwCas13a by incubating overnight with unsupplemented lysis buffer, SUMO protease (Fisher Scientific), and NP-40. Cleaved LwCas13a was collected by elution from column and further purified by ion exchange chromatography (5 ml HiTrap SP HP, GE Healthcare Life Sciences) followed by size exclusion chromatography (Superdex 200 Increase 10/300 GL (GE Healthcare Life Sciences). LwCas13a was stored at -80˚C in storage buffer (0.05 M Tris-HCl pH 7.5, 0.6 M NaCl, 2 mM DTT, and 5% glycerol).

## Sequence analysis and design of RT-RPA primers, crRNAs and synthetic CCHFV RNAs

The S segments of CCHFV strains representing each of the different genetic clades or sub-clades (Table 1) were aligned using the Clustal W algorithm in the DNASTAR Lasergene software to identify conserved regions as potential RPA amplicons including binding sites by RPA primers and crRNAs. The sequences and genomic locations of RPA primers, crRNAs and synthetic positive control CCHFV RNAs corresponding to the RPA amplicons are listed in Table 2. These sequences were synthesized by Integrated DNA Technologies or GenScript.

## Production of the S segment RNA genome from CCHFVs and CCHFV-related viruses by in vitro transcription

The negative-sense S segment sequence from each of the CCHFV strains (Table 1) and CCHFV-related viruses (Table 3) was custom cloned into pET-28c(+) downstream of the T7 promoter by GenScript. The sequences and cloning strategies are detailed in Figs A and B in S1 File. For in vitro transcription, 1 µg linearized construct was purified using the QIAquick Gel Extraction Kit (Qiagen, 28704) and eluted with 30 µl nuclease-free water (ThermoFisher Scientific, 750024). 8 µl of the eluted DNA was then in vitro transcribed using the HiScribe T7 High Yield RNA Synthesis Kit (New England BioLabs, E2040S) in a 20 µl reaction also containing 2 µl of each of these components: 10X Reaction Buffer, 100 mM ATP, CTP, GTP and UTP, T7 RNA Polymerase Mix. The reaction was incubated at 37˚C for two hours or, to increase RNA yield, overnight (about 16 hours). The removal of template DNA was performed by addition of 70 µl nuclease-free water, 10 µl of 10X DNase I Buffer (New England BioLabs, part of M0303L), and 2 µl of RNase-free DNase I (New England BioLabs, M0303L), followed by mixing and incubation for 15 minutes at 37˚C. The RNA was then purified using the RNeasy Mini Kit (Qiagen, 74104) and quantified on a NanoDrop One spectrophotometer (ThermoFisher Scientific, ND-ONE-W4).

## Limit of detection (LoD)

An assay cutoff of negative threshold was established as mean plus 3 standard deviations of no-template controls (NTCs), based on 10 NTC replicates. A relative fluorescence unit (RFU) value above the negative threshold is considered CCHFV positive. The LoD was defined as the lowest concentration at which 100% of the repeats were positive.

## Viruses, cells, and production of cell culture-derived viral RNAs

CCHFV IbAr10200 was a kind gift from David Safronetz (Public Health Agency of Canada) and Heinz Feldmann (Then Public Health Agency of Canada) [43,44]. SW13 (Scott and White

**Table 1. Clade-representing CCHFV sequences.**

| Short name of CCHFV sequence (the S segment) | GenBank accession | Strain name | Clade (Subclade) by Carroll et al. 2010* | Clade by Atkinson et al. 2012** |
|---|---|---|---|---|
| 1.ArD | DQ211640 | ArD15786 | I | Africa 1 |
| 2.UG | DQ211650 | UG3010 | II | Africa 2 |
| 3.IbAr | U88410 | IbAr 10200 | III | Africa 3 |
| 4.1Bagh | AJ538196 | Baghdad-12 | IV(1) | Asia 1 |
| 4.2CHN | AJ010649 | China 8402 | IV(2) | Asia 2 |
| 5.Hoti | DQ133507 | Kosova Hoti | V | Europe 1 |
| 6.AP | U04958 | AP92 | VI | Europe 2 |

* Carroll et al. Mol Phylogenet Evol. 2010 Jun;55(3):1103–10. PMID: 20074652.

** Atkinson et al. Euro Surveill. 2012 Nov 29;17(48). PMID: 23218389.

**Table 2. Primers, crRNAs and chemically synthesized CCHFV RNA fragments.**

| CRISPR set No. | Amplicon location | crRNA location | Type | Name / Abbreviation | Sequence (5'-3') |
|---|---|---|---|---|---|
| 1 | 1–120 | 3–30 | Synthetic RNA target | Target_RNA_1–120 | UUUUUAAACUCCCUCAAACCAUUUGUUCAUCUCAUCCUUGCUGUCUGUGUGUUUUCCAUUUUGCAUUUUUGCAGACUACUCAAGAGAACACUGUGGGCGUAAGCGGCGACGUGUUCUUUUGAGA |
| | | | RT-RPA forward primer | RPA_T7_1-37f | GAAATTAATACGACTCACTATAGGGTCTCAAAGAAACACGTGCCGCTTACGCCCACAGTGTT |
| | | | RT-RPA reverse primer | RPA_93-120r | TTTTTAAACTCCTCAAACCATTTGTTCA |
| | | | crRNA | crRNA_3–30 | GAUUUAGACUACACCCAAAAACGAGGGGACUAAAACUGGGCGUAAAACUGGGCGCACGUGUUCUUUGA |
| 2 | 641–723 | 669–696 | Synthetic RNA target | Target_RNA_641–723 | GAUUUGUUUGAUGUCCCCCAAGGUGGAUUGAAGGCCAUGAGUGUACUGCCUUUGACAAACUCCCUGCACCACUCCACAUGUUC |
| | | | Synthetic RNA target | Target_IND | GACUUGGUUGAUGUCCCCCAGGGUGGGUUGAAAGCCAUUAUGUACUGCCUUUAACAAAUUCCCUACACCACUCCACAUGUUC |
| | | | Synthetic RNA target | Target_UAE | GACUUGGUUGAUGUCCCCCAAGGUGGGUUAAAAGCCAUUAUGUACUGCCUUUGACAAAUUCCCUCACACCACUCCACAUGCUC |
| | | | Synthetic RNA target | Target_Rec | GACUUGGUUGAUGUCCCCCAAGGUGGGUUAAAAGCCAUUAUAUAUUUGCCCUGACAAAUUCCCUACACCACUCCACAUGUC |
| | | | RT-RPA forward primer | RPA_T7_641-672f | GAAATTAATACGACTCACTATAGGGGAACATGTGGAGTGGTGCAGGAGTTTGTCAA |
| | | | RT-RPA reverse primer | RPA_689-723r | GATTTGTTGATGTCCCCCAAGGTGGATTGAAGGC |
| | | | RT-RPA forward primer | RPA_T7_641-672f_D | GAAATTAATACGACTCACTATAGGGGAACATGTGGAGTGGTGCAGGARTTTGTCAA |
| | | | RT-RPA reverse primer | RPA_689-723r_D | GATTTGTTGATGTCCCCCAAGGTGGRTTGAARGC |
| | | | crRNA | crRNA_669–696 / crRNA_Deg | GAUUUAGACUACACCCAAAAACGAGGGGACUAAAACUGGAARGGCCAUDAURUAYUURKCYUUGA |
| | | | crRNA | crRNA_669–696_5.Hoti / crRNA_Con | GAUUUAGACUACACCCAAAAACGAGGGGACUAAAACUGGAAGGCCAUGAGUGUACUUGCGCUUUGA |

Location numbes are based on the S segment of the CCHFV genome (positive sense sequence, GenBank accession number: DQ133507) from the Kosovo Hoti strain (standard reference strain in this study). Non-CCHFV sequences are shown in italics, including a T7 promoter sequence in forward primers and a DR sequence in crRNAs. Degenerate nucleotides are highlighted in bold. Target_IND and Target_UAE are based on GenBank accession numbers MH396666 and MF289419. Target_Rec is based on recombination between DQ211640 and MF289419. In the last two crRNAs, "Deg" and "Con" represent degenerate and constant spacer sequences.

**Table 3. CCHFV-related viruses (Family Bunyaviridae).**

| Virus | Genus | GenBank accession (the S segment) |
|---|---|---|
| Hazara virus | Nairovirus | NC_038711 |
| Nairobi sheep disease virus | Nairovirus | AF504294 |
| Dugbe virus | Nairovirus | AF434164 |
| Oropouche virus | Orthobunyavirus | KP052852 |
| Hantaan virus | Hantavirus | M14626 |
| Rift Valley fever virus | Phlebovirus | MH175205 |

No. 13) cells were kindly provided by Darwyn Kobasa (Public Health Agency of Canada) [45,46]. The cells were seeded in Leibovitz's L-15 medium (ThermoFisher Scientific, 11415–0964), supplemented with 10% Fetal Bovine Serum (FBS). Growth was maintained at 37°C without $CO_2$. Once cells reached ~80% confluence, they were infected with the CCHFV at an MOI (multiplicity of infection) of 0.1. At 3 dpi (days post-infection), when significant CPE (cytopathic effect) was observed, culture supernatants were harvested and inactivated by the TriPure Isolation Reagent (Millipore Sigma, 11667165001) by mixing 100 μl supernatants with 900 μl TriPure in a 1.5 ml Eppendorf tube. Purification of CCHFV RNA was then performed using the RNeasy Mini Kit (Qiagen, 74104). The inactivated supernatants were mixed with the RLT buffer at a ratio of 100 μl: 350 μl. To concentrate RNA yield, a 2-ml input of the inactivated supernatants was used in one spin column.

A Rift Valley fever virus isolate from the 2006–2007 Kenyan outbreak (RVFV-UAP; GenBank accession number MH175203, MH175204, or MH175205) was alternately passaged between Vero E6 cells and C6/36 cells, as previously described [47,48]. TriPure-inactivated samples as prepared above were extracted for viral RNA using the MagMAX CORE Nucleic Acid Purification Kit (ThermoFisher Scientific, A32700) on a KingFisher magnetic particle processor (ThermoFisher Scientific, A31508).

## Results

### Sequence divergence in CCHFV

We chose to target the S segment of CCHFV for testing the degenerate sequence-based CRISPR diagnostic strategy. It is the most conserved and the most often targeted region by previous diagnostic development attempt among CCHFV genomic segments. Previous studies had revealed a 20% sequence divergence in the S segment [29]. A similar number, 20.152%, was found in our confirmatory phylogenetic analysis (S2 File) and multiple alignment analysis (S3 File), based on all CCHFV isolates that we found with a complete S segment sequence described in GenBank. In addition, extensive sequence variations were visualized by a Shannon's entropy plot across individual nucleotide positons of the S segment sequences (Fig C in S1 File). These results confirmed the suitability of CCHFV to serve as a representative variable viral target.

### Assay design and screening of CRISPR sets

The degenerate sequence-based CRISPR diagnostic strategy was tested in the format of a Cas13a-mediated SHERLOCK assay targeting the S segment of the CCHFV genome. The assay consists of two steps (Fig 1). In the first step, viral RNA is converted to DNA by reverse transcription (RT) and amplified by RPA. A pair of RPA primers, one with a T7 promoter sequence, define the amplicon. In the second step, the DNA amplicon is T7-transcribed and amplified into RNA. The resulting target RNA is then recognized by a crRNA with a

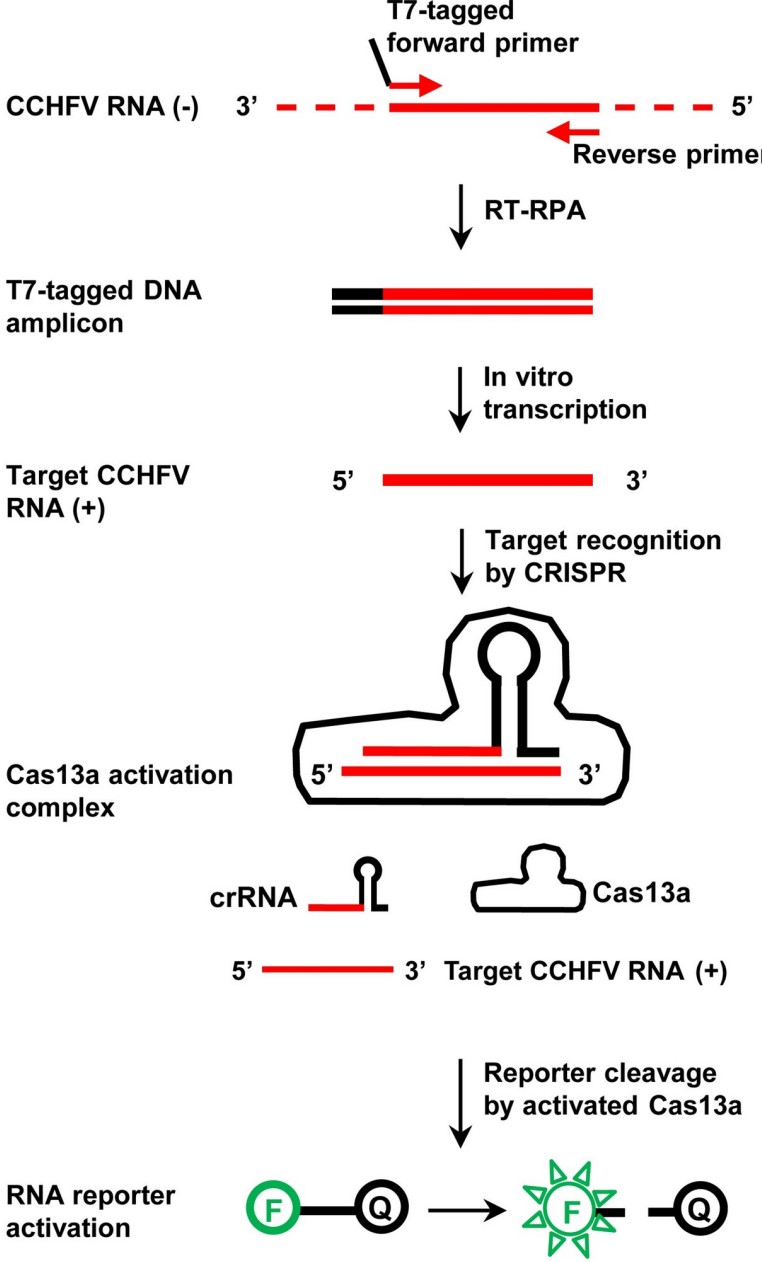

**Fig 1. The CRISPR/Cas13a-based diagnostic assay (SHERLOCK) for CCHFV detection.** The polarity of an RNA strand is labelled for positive sense (+) or negative sense (-). In the RNA reporter, "F" and "Q" mean fluorophore and quencher, respectively.

degenerate spacer sequence, which corresponds to a region within the RPA amplicon. Target recognition leads to the activation of Cas13a, cleavage of RNA reporters and release of fluorescent signals.

A number of traditional CCHFV strains were involved in this study and are listed in Table 1, representing each of the different genetic clades or sub-clades. Three major designation systems of CCHFV clades [29] have been employed by different researchers, including Carroll et al [49], Atkinson et al [50] and Mild et al [39]. According to these, six or seven CCHFV clades have been identified. The disagreement appears to result from minor

preferences in the views of closely-related branches of phylogenetic trees, but does not reflect any critical difference [29]. We adopted the six-clade system by Carroll et al, but divided Clade IV into Sub-clades 1 and 2, represented as Clade IV(1) and Clade IV(2) and corresponding to Clades Asia 1 and 2 in the system by Atkinson et al, respectively (Table 1).

The S segments of CCHFV strains, available from GenBank as sense (positive-polarity) sequences, were aligned to identify the most conserved regions as potential binding sites by any candidate set of RPA primers and crRNAs (referred to as "CRISPR set" from now on). Two lead CRISPR sets are displayed in Fig 2 and Table 2. In CRISPR set 1 (Fig 2A), the RPA amplicon involved the 5' region of the S segment between nucleotide positions 1 and 120. Unless otherwise noted, all nucleotide position numbers to be described later are defined based on the Kosovo Hoti strain (the reference strain in this study), while the same homologous regions may have different position numbers in different viral strains due to sequence shifts caused by insertions or deletions. The amplicon of CRISPR set 1 overlaps with the region previously targeted by a CCHFV RPA [51]. Out of the entire S segment, the first 30 nucleotides are the most conserved, with no more than one variable position. We took this chance to design a traditional type of crRNA for CRISPR set 1, with a constant spacer sequence targeting positions 3–30 (crRNA_3–30). In CRISPR set 2 (Fig 2B), the RPA amplicon involved positions 641–723 and the crRNA targeted positions 669–696 (crRNA_669–696, or crRNA_Deg). These overlap with the amplicon and Taqman probe from two previous CCHFV qPCR diagnostics [52,53]. Several mutation hotspots were identified in the crRNA target sequence, while the other positions remain constant (Fig 2B). The mutation spots included positions 5, 8, 11, 14, 17 and 23 of the crRNA target sequence (CCHFV sense sequence). This pattern was found to be consistent among nearly all CCHFV strains by Nucleotide BLAST (National Center for Biotechnology Information) search. A few exceptions will be described later. To cover all mutations in these hotspots, we applied the degenerate sequence-based crRNA design to CRISPR set 2. Mixed nucleotide types corresponding to variations at each position were introduced into the spacer sequence of crRNA_669–696 during its chemical synthesis (Fig 2B and Table 2).

We conducted a pilot test and comparison of the two crRNA designs (Fig 2C and 2D). In this context, the RPA primers of both the CRISPR sets had a constant sequence, based on the Hoti strain (Table 2). These included primers RPA_T7_1-37f and RPA_93-120r for CRISPR set 1 and primers RPA_T7_641-672f and RPA_689-723r for CRISPR set 2. The consideration was that RPA is well tolerant of mismatches and the small number of mutations in the primer binding sites was not expected to significantly inhibit the RPA reaction [54]. The CRISPR sets were initially characterized for detection of chemically synthesized CCHFV RNA molecules (Fig 2C and 2D and Table 2). These corresponded to positions 1–120 (Target_RNA_1–120) and positions 641–723 (Target_RNA_641–723) of the S segment from the Hoti strain, respectively (Table 2). CRISPR set 1 showed signal amplification in all tested reactions, in the presence of the intended positive control template (Target_RNA_1–120), the negative control template (Target_RNA_641–723), or non-template control (NTC, H2O) (Fig 2C). The false positive results may arise from the overlap between the T7 promoter-tagged RPA primer (RPA_T7_1-37f) with the spacer sequence of crRNA_3–30 (Fig 2A) [19]. The overlap, however, was unavoidable as the primer and spacer were both located to the very end of the 5' region of the S segment (Fig 2A). CRISPR set 2 detected the positive control template (Target_RNA-741-723) specifically, but not the negative control template (Target_RNA_1–120) or non-template control (NTC, H2O), demonstrating that a crRNA with a degenerate spacer sequence can work effectively to recognize a specific target sequence and activate the CRISPR--Cas13a system (Fig 2D). CRISPR set 2 was then used to represent the degenerate sequence-based CRISPR diagnostic for further evaluation.

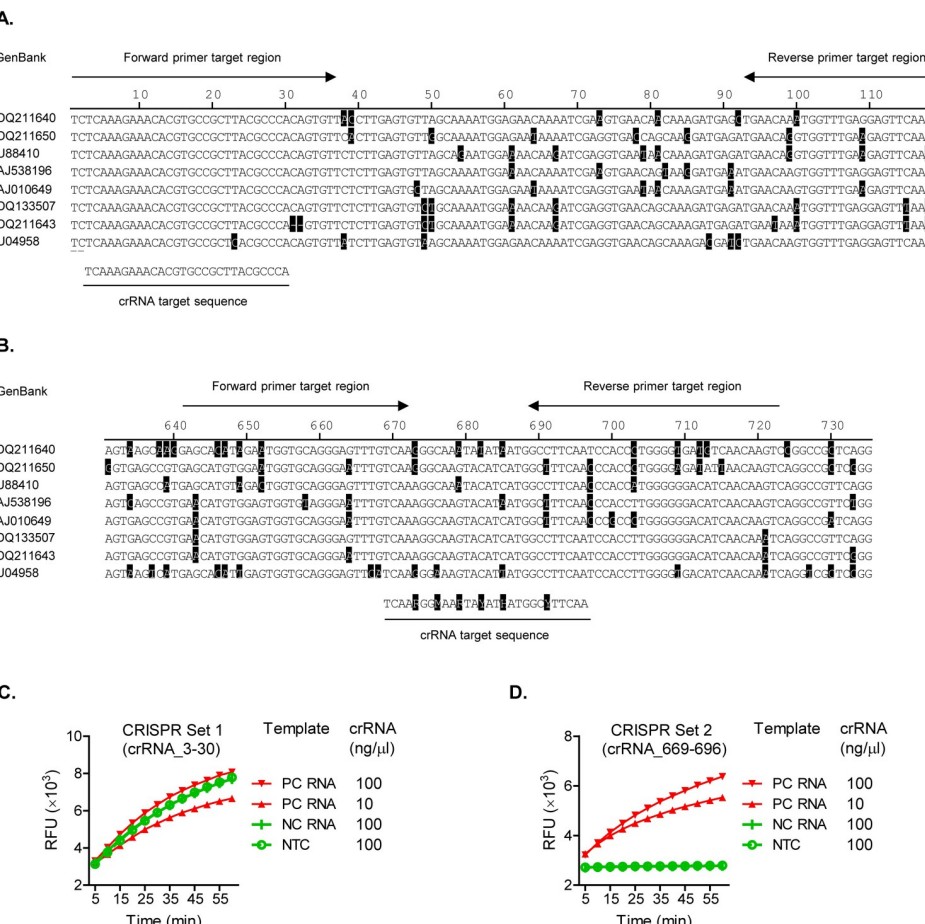

**Fig 2. Design and function test of CCHFV CRISPR sets. A and B.** Sequences and genomic locations targeted by CCHFV CRISPR sets. The S segment sequences of CCHFV strains, labelled with GenBank accession numbers, were aligned at the assay design regions for CRISPR set 1 (in Panel A) and CRISPR set 2 (in Panel B). Nucleotides differing from the majority are highlighted in dark shades. Sequences and locations targeted by RT-RPA primers and crRNAs are indicated, based on the positive-sense sequence of the S segment. The CCHFV strains, representing each of all the seven clades/sub-clades, are detailed in Table 1, except that a Clade V variant, DQ211643, with a two-nucleotide deletion at genomic positions 31 and 32 is included only in this figure. Note that degenerate nucleotides were introduced into the crRNA of CRISPR set 2, to cover variations at each position in the targets, and are displayed with the standard codes by International Union of Pure and Applied Chemistry (IUPAC). **C and D.** Specific signal amplification by CRISPR Set 2, but not CRISPR set 1. The CRISPR sets were tested in the CRISPR/Cas13a-based SHERLOCK assay, in the presence of chemically synthesized CCHFV RNA fragments and different concentrations of stock crRNAs. Sequences of RT-RPA primers, crRNAs and synthetic RNA templates are listed in Table 2. Amplification plots show relative fluorescent units (RFU) at indicated time points in the T7-Cas13a reaction. **C.** Test of CRISPR set 1 (with constant crRNA spacer). PC RNA (positive control RNA, specific target) = Target_RNA_1–120; NC RNA (negative control RNA, non-specific target) = Target RNA_641–723. NTC (no-template control) = H2O. Graph represents five independent experiments showing similar patterns. **D.** Test of CRISPR set 2 (with degenerate crRNA spacer). PC RNA = Target RNA_641–723; NC RNA = Target_RNA_1–120. NTC = H2O. Graph represents four independent experiments showing similar patterns.

## Sensitive and rapid detection of all CCHFV clades

We next applied the degenerate sequence-based CRISPR diagnostic to the detection of different CCHFV clades. Previous RPA studies suggested that mismatches near the 3' end of primers more likely affect productivity [54]. To maximize the amplification of CCHFV variants, we updated our RPA primers by introducing degenerate nucleotides into a few positions close to their 3' end corresponding to variations in the CCHFV targets. These included position 664

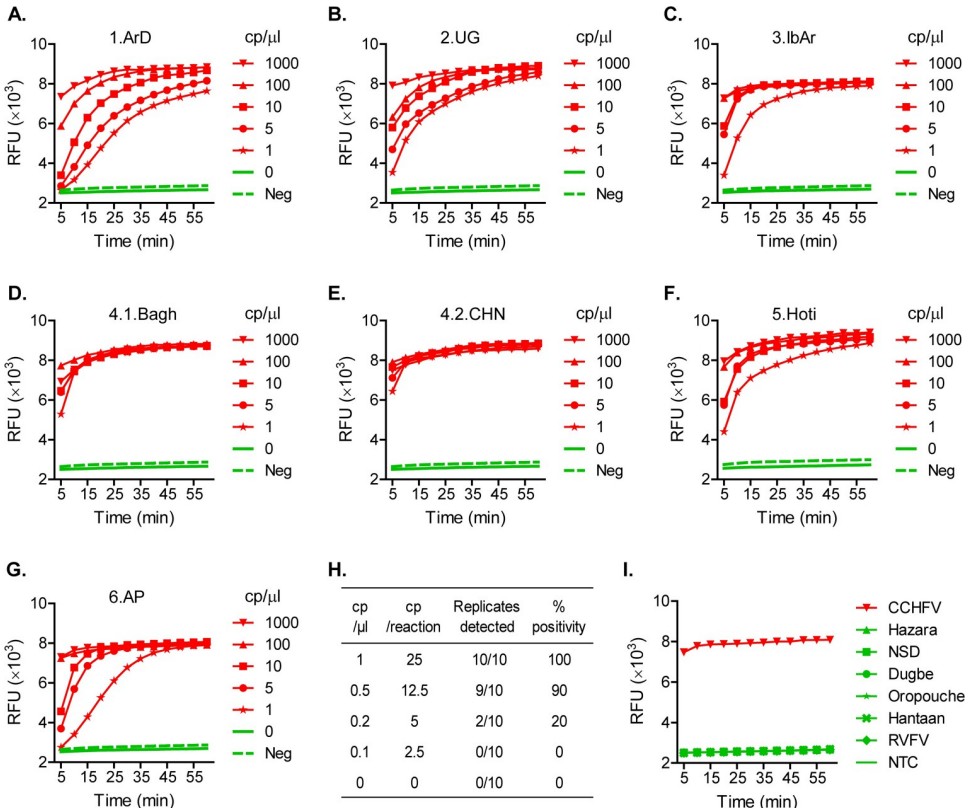

**Fig 3. The degenerate sequence-based CRISPR diagnostic detects CCHFV in a sensitive, specific and rapid manner. A–G.** Detection of all CCHFV clades/sub-clades. The S segment genomic RNAs representing different CCHFV clades/sub-clades were produced by in vitro transcription, serially diluted and tested for detection by the degenerate CRISPR set (CRISPR set 2). Amplification plots show relative fluorescent units (RFU) at indicated time points in the T7-Cas13a reaction. RNA template concentrations in the T7-Cas13a reaction are shown as copies/μl (cp/μl). Red curves: CCHFV RNA-containing samples; green solid curve (0 cp/μl): NTC (no-template control, H2O); and green dash curve (Neg): the negative threshold = Mean + 3X standard deviation of NTCs, calculated based on 12 NTC replicates. Any RFU value above the negative threshold is considered CCHFV positive. Graphs represent three independent experiments showing similar patterns. **H.** Limit of detection (LoD). The S segment genomic RNA from the Hoti strain was produced by in vitro transcription and tested at the indicated concentrations for detection by the degenerate CRISPR set. Positivity% values are shown based on 10 replicates per target concentration type. **I.** The degenerate sequence-based CRISPR diagnostic is specific for CCHFV, with no cross-reactivity against closely related viruses. RNAs from CCHFV and related viruses were tested for detection by the degenerate CRISPR set. NSD: Nairobi sheep disease virus; RVFV: Rift Valley fever virus; and NTC: no-template control (H2O). CCHFV and Rift Valley fever virus RNAs were extracted from cell culture-derived viruses. Other viral RNAs were the S segment genomic RNAs produced by in vitro transcription. RNA template concentrations in the T7-Cas13a reaction were all 40 pg/μl. Amplification plots show relative fluorescent units (RFU) at indicated time points in the T7-Cas13a reaction and represent three independent experiments showing similar patterns.

(defined by that in the target sequence, Hoti strain) in primer RPA_T7_641-672f_D and positions 691 and 697 in primer RPA_689-723r_D (Fig 2B and Table 2).

To obtain CCHFV RNA targets, the full-length S segment sequences (in negative sense to mimic the natural CCHFV RNA genome), representing each of the seven CCHFV clades/sub-clades, respectively, were cloned into a T7 promoter-containing vector, pET-28c(+), for in vitro transcription (Table 1, and Fig A in S1 File). The resulting RNA samples were serially diluted and tested using the updated CRISPR set 2. The degenerate sequence-based CRISPR diagnostic detected all the CCHFV clades/sub-clades in a highly sensitive and rapid fashion (Fig 3A–3G, and Fig D in S1 File). In all the clades/sub-clades, the assay demonstrated strong signal amplification from target RNA concentrations (in the final T7-Cas13a reaction) at or

above 10 copies (cp)/μl within five minutes into the T7-Cas13a step, or within 30 minutes of the assay start (considering the earlier, 25-minute RT-RPA step) (Fig 3A–3G). In addition, target RNA at 1 cp/μl from any clade was statistically confirmed to result in significant signal amplification within 35–40 minutes of the assay start (Fig D in S1 File). Moreover, we tested the limit of detection (LoD) using the CCHFV Hoti strain as a representative target (Fig 3H). The assay showed a positivity rate of 100% (10/10 replicates) with 1 cp/μl of target concentration (or 25 cp/reaction, based on a 25 μl final T7-Cas13a reaction), and 90% (9/10) with 0.5 cp/μl (or 12.5 cp/reaction). The positivity rate dropped sharply when lower target concentrations were tested. In turn, the LoD of the assay confidently fell in the range around 1 cp/μl (Fig 3H).

## Specificity against CCHFV-related viruses

To evaluate the specificity of the degenerate sequence-based CRISPR diagnostic, the sequences of the RPA primers and crRNA spacer were searched by BLAST and confirmed to have no significantly high similarity to any non-CCHFV sequences. Experimentally, the assay specificity was tested against a panel of viruses closely related to CCHFV, including Hazara virus, Nairobi sheep disease virus, Dugbe virus, Oropouche virus, Hantaan virus and Rift Valley fever virus [55,56] (Table 3, Fig 3I, and Figs B and E in S1 File). These viruses have been used in negative viral panel testing to validate the specificity of a commonly used CCHFV Real-Time RT-PCR assay [57]. Negative-sense target RNA samples from all these viruses except Rift Valley fever virus were prepared through in vitro transcription based on cloning of their S segment sequences into the pET-28c(+) vector (Fig B in S1 File). Rift Valley fever virus RNA was extracted from virus stocks propagated in cell culture. For comparison, we also included cell culture–derived viral RNA from CCHFV (IbAr10200 strain) in the experiments. Consistent with the bioinformatic prediction, all the CCHFV-related viruses produced negative results, while in sharp contrast CCHFV generated strong amplification signals (Fig 3I, and Fig E in S1 File). These data indicated that the degenerate sequence-based CRISPR diagnostic for CCHFV is highly specific. In addition, the assay was confirmed to detect not only CCHFV RNA produced by chemical synthesis and enzyme-based in vitro transcription, but also RNA produced biologically from the real virus. It should be noted that the assay has not yet been tested using CCHF clinical samples, as we do not have access to this sample type. In clinical samples human genomic DNA is a major known contaminant that often leads to false positive results in nucleic acid amplification assays [58,59]. To examine the effect of this potential contaminant from patient samples, we spiked commercially available human genomic DNA in CCHFV RNA samples at concentrations ranging from 2 pg/μl to 20 ng/μl for CRISPR testing. The resulting signal amplification patterns were similar comparing the conditions with and without added DNA, and demonstrated that human genomic DNA contamination does not cause any significant false positive signal in the degenerate sequence-based CRISPR assay for CCHFV (Fig F in S1 File).

## Detection of more divergent mutant sequences

Finally, we asked if the degenerate sequence-based CRISPR diagnostic is capable of detecting emerging viral variants with new and more complex mutations. Our degenerate crRNA spacer sequence covers six mutation hotspots, which were identified using traditional CCHFV strains (Fig 2B). These are located at positions 5, 8, 11, 14, 17 and 23 of the crRNA target sequence (CCHFV sense sequence). We further considered that new mutation types could possibly develop in newer and more divergent viral isolates. An extensive BLAST search confirmed that all CCHFV variants except a few are fully covered by the degenerate crRNA sequence in terms of both the positions and nucleotide types of mutations. The exceptional CCHFV

variants can be divided into two types. Type I variants are characterized by a new mutation, located at position 2 of the crRNA target sequence (CCHFV sense sequence). At this site, the commonly seen nucleotide in traditional CCHFV isolates, C (Cytosine), is replaced by T (Thymine) (Fig 4A). Type II variants are characterized by a new mutation at position 26, which is also C to T (Fig 4B). The two variant types can be represented, respectively, by a viral isolate identified in 2018 from India (IND) with a GenBank accession number MH396666, and by a viral isolate identified in 2017 from United Arab Emirates (UAE) with a GenBank accession number MF289419 (Table 2 and Fig 4A and 4B). Phylogenetic analysis showed that these isolates belong to Clade IV(1) (Fig G in S1 File).

As shown in Fig 4A and 4B, both the IND and UAE variants were effectively detected by crRNA_669–696 (labelled as crRNA_Deg in the figure, to better distinguish from a constant-spacer crRNA), demonstrating a success of the degenerate sequence-based CRISPR diagnostic in detecting newer and more divergent CCHFV mutants. For comparison, in the same experiments we tested a crRNA with a constant spacer sequence derived from the Hoti strain, crRNA_669–696_5.Hoti (labelled as crRNA_Con in the figure, to better distinguish from a degenerate-spacer crRNA). The constant-spacer crRNA also detected the IND and UAE variants successfully (Fig 4A and 4B). It is noteworthy that this detection was achieved in the presence of three mismatches found between the constant spacer and the IND and UAE target sequences (Fig 4A and 4B). While a limited mismatch tolerance by CRISPR could be implied by previous cases where as few as two mismatches inhibited CRISPR target recognition [19], the current finding suggested a more decent degree of tolerance, and that the effect of mismatches may be context-dependent. Additionally, in this comparative test the degenerate and constant sequence-based CRISPR sets showed similar signal amplification patterns, indicating that the degenerate sequence-based CRISPR design can maintain a level of assay activity similar to that of a traditional CRISPR.

Previous studies suggested long distance virus transfer and genetic recombination as part of the factors contributing to CCHFV diversity and potential future emergence of new variants [29]. In Fig 4C, a recombinant CCHFV mutant, consisting of sequences found in Clades I and IV, was effectively detected by the degenerate but not the constant sequence-based crRNA. This result demonstrated that the degenerate sequence-based CRISPR has an enhanced capacity in tolerating more complex, higher degree mismatches.

Taken together, the findings from this proof-of-concept study demonstrated that the degenerate sequence-based CRISPR design strategy can be used to develop functional, sensitive, specific and rapid CRISPR diagnostics, as a useful tool for detecting highly variable viral pathogens.

## Discussion

This study explored novel opportunities in the development of CRISPR diagnostics, for effective detection of highly variable targets, especially RNA viruses such as CCHFV. It was prompted by previous observations that CRISPR target recognition can be abolished by mismatches between the CRISPR spacer and target sequences. In some instances, target recognition can fail with as few as two mismatches, and based on a Zika CRISPR assay which was successful at distinguishing variants differing by one nucleotide, it is suggested that an achievable single-nucleotide resolution of detection could be obtained [19]. While these data tend to imply a stringent intolerance of mismatches by CRISPR, however, it has not been clearly understood how specific types of mismatches affect CRISPR detection. Therefore, when designing a CRISPR diagnostic test, an uncertain effect of different types of mismatches should be taken into consideration.

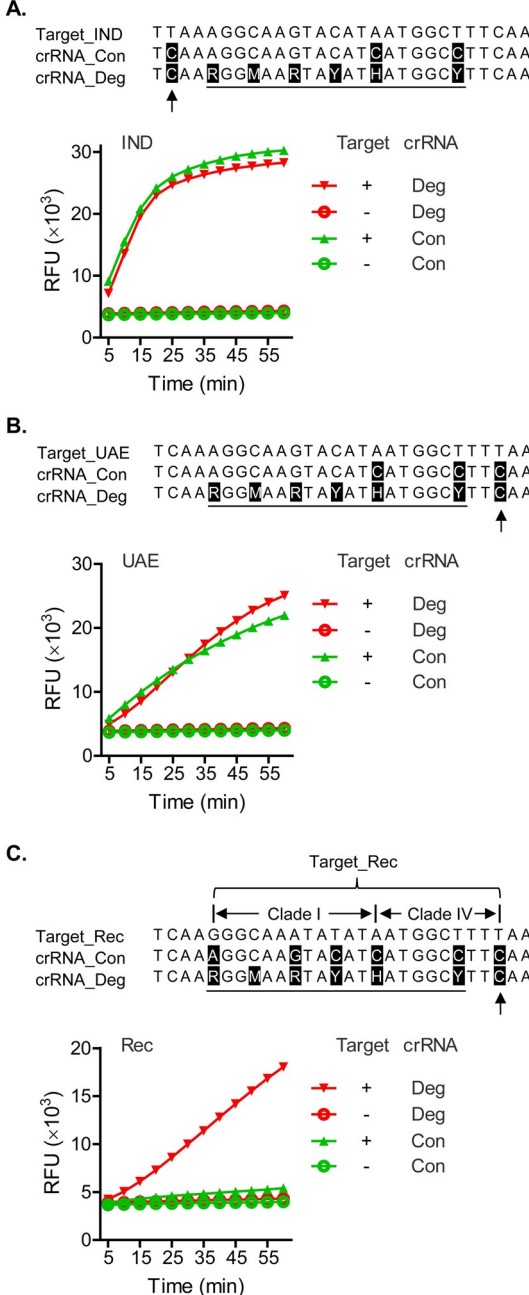

**Fig 4. The degenerate sequence-based CRISPR diagnostic is capable of detecting emerging CCHFV variants with new and more complex mutations.** Chemically synthesized CCHFV RNA targets were tested for detection at 1 cp/μl by the degenerate sequence-based CRISPR set, using crRNA_669–696 (crRNA_Deg), as in Fig 3. A constant sequence-based crRNA, derived from the Hoti strain, crRNA_669–696_5.Hoti (crRNA_Con), was compared to the degenerate sequence-based crRNA in the same context. The targeted CCHFV variants were isolates from India (IND, 2018) and United Arab Emirates (UAE, 2017) in Panels A and B, respectively, as well as a recombinant (Rec) between Clades I and IV. Sequences of synthetic RNA templates, RT-RPA primers and crRNAs and GenBank accession numbers of CCHFV strains are provided in Table 2. Sequence of the IND, UAE or Rec variant targeted by CRISPR was each aligned with the degenerate and constant spacer sequences of the crRNAs (all as CCHFV sense sequences). Each nucleotide in the crRNA spacers that differs from that in the targeted CCHFV variant is shaded. In crRNA_Deg, the region is underlined where each degenerate nucleotide covers its counterpart in the target sequence, meaning that one of the mixed nucleotide types represented by the degenerate code matches the counterpart nucleotide in the target. Arrow indicates the nucleotide in crRNA_Deg that does not cover/match the counterpart nucleotide in the target. In Panel C, regions in the target sequence with mutations characteristic of Clade I and Clade II are demarcated,

respectively. Each amplification plot shows relative fluorescent units (RFU) at indicated time points in the T7-Cas13a reaction, in the presence (+) or absence (-) of the target CCHFV RNA, using an indicated crRNA. Data are represented as mean of three technical replicates.

When a traditional constant spacer-based crRNA was used to detect the more divergent CCHFV strains, IND and UAE, our data demonstrated that three mismatches (more than two) were well tolerated. The discrepancy between the current and previous findings may be explained by the possibility that the effect of mismatches depends not only on their number, but also other factors such as their locations within the spacer and the context of their neighboring sequences. Further studies are required to better characterize these hypotheses. Furthermore, constant spacer-based CRISPR diagnostics should have an increased capacity to cover variable viral targets, if the targeted genomic region is well conserved and has limited types of mutations that fall within the range of tolerance by the crRNA used. It is conceivable that the tolerance of some specific types of mismatches between a constant spacer and a mutant target sequence, such as those with number and locations that have not been reported previously (and thus with uncertain effect), may not be confidently predictable during the assay design but can only be determined after extensive experimental testing and optimization. This may often involve the designing and screening of multiple CRISPR sets targeting different genomic regions.

As a new alternative to the traditional constant-spacer CRISPR, the degenerate-spacer CRISPR represents a proof of concept to build "certainty" into the assay design. Since mixed nucleotide types matching all viral mutations of interest can be directly included in the crRNA design, achieving a coverage of entirety in the very beginning of the assay development process is made possible. This advantage is anticipated to award a simpler and straight forward path leading to the establishment of a successful CRISPR set that covers all mutants of a variable viral target.

Our evaluations demonstrated that the degenerate sequence-based CRISPR diagnostic for CCHFV is functional and has maintained the advantages of traditional CRISPR diagnostics (isothermal, sensitive, specific and rapid). First, degenerate sequences have not been previously tested in CRISPR. A potential issue would have been that a degenerate crRNA is actually a mixture of crRNA species with different spacer sequences. Some of them are expected to match the target sequence well enough in order to activate the Cas enzyme, while the others are not, due to mismatches that are not tolerated. These non-functional crRNA species would potentially compete for binding to the Cas enzyme and target RNA, impairing CRISPR activation. Despite this, the degenerate sequence-based CRISPR-Cas13a assay was able to detect all CCHFV clades effectively. It demonstrated a sensitivity at 1 cp/µl level, which is typical of traditional CRISPR diagnostics. In a side-by-side comparison, the degenerate-spacer CRISPR and a traditional constant-spacer CRISPR showed nearly identical signal amplification patterns when detecting the IND and UAE strains. The maintenance of functionality with high sensitivity in the degenerate-spacer CRISPR may be explained by the possibility that the binding of crRNA to the Cas enzyme and target RNA is dynamic and reversible, and the functional crRNA species outcompete the others due to a larger number and a better matching of their spacer sequences to the target.

To obtain additional information regarding the limit of detection (LoD) of the assay, it is noteworthy that real-time RT-PCR has so far been the most widely used molecular method to detect CCHFV genomic sequences. A number of commercial and in-house assays have been developed in recent years [60]. The LoDs of these assays largely fall in a range of 5–1000 cp/reaction and are frequently seen around 10 cp/reaction [53,57,61,62]. The LoD of the

degenerate-sequence CRISPR assay, at roughly 25 cp/reaction, appeared to be comparable to those of real-time RT-PCR assays.

The degenerate-spacer CRISPR assay has also maintained high specificity in the presence of degenerate nucleotides. We had considered several factors that should together contribute to the specificity. The crRNA spacer-targeted region was among the most conserved in the CCHFV genome. As a result, a conserved backbone consisting of positions with a constant nucleotide occupant dominated the targeted region. These in combination with less conserved but still CCHFV-specific sequences from the variable positions contribute to crRNA specificity. Finally, a combinatorial, enhanced specificity of the whole CRISPR diagnostic was determined by the crRNA and RPA primers together.

As far as the run time is concerned, the assay took 35–40 min (including 25 min RPA and 5–15 min CRISPR/Cas13a) to detect 1cp/µl CCHFV RNA. This is typical of traditional CRISPR and shorter than the run time of PCR-based diagnostics, which normally take no less than one hour. For example, a popular commercial kit, RealStar CCHFV RT-PCR Kit 1.0 (Altona-Diagnostics), runs about 95–100 min (using the supplier's PCR protocol on an Applied Biosystems 7500 Real-Time PCR System). Recent advances in CRISPR diagnostics (further discussed later) such as combining the RPA and CRISPR/Cas13a steps into a one-pot format, if adopted in future development, are expected to make the assay more rapid.

The degenerate-spacer CRISPR also demonstrated additional advantages. It effectively detected a recombinant mutant sequence derived from CCHFV clades I and IV, which a constant-spacer CRISPR failed to, indicating a higher capacity of mutation tolerance. Finally, based on its design principle, the degenerate-spacer CRISPR has the flexibility to be quickly adapted to emerging mutations by simply adding degenerate nucleotides to the spacer corresponding to the new mutations. This is an extension of its advantage in assay design, on top of the built-in confidence in covering previously known mutation types as discussed above. Taking these advantages together, the degenerate-spacer CRISPR holds great potential to serve as a new alternative to the traditional CRISPR and a useful tool of choice for detecting highly variable viral pathogens.

The CRISPR assay developed so far in this study is not yet intended for use in clinical diagnosis of CCHFV infection. The study had been designed, on a proof-of-principle basis, to focus on an initial test of the feasibility to use a degenerate sequence-based CRISPR against highly variable targets. While the data presented here have fulfilled that scope, the assay has not been fully validated using clinical samples, to which we currently have no access, although several other types of CCHFV RNA sources have been involved, including chemical synthesis, enzyme-based in vitro transcription and cell culture propagated, whole viruses. Contaminants potentially carried from clinical samples could affect the assay performance. A major one of these, human genomic DNA, is commonly known to interfere with the specificity of nucleic acid tests, causing false positive signals. By spiking it into CCHFV RNA samples, we validated that it does not impact the specificity of the CCHFV CRISPR assay.

Moreover, while considerably functional, sensitive, specific and fast, the current assay is not intended as an ultimate end product having reached the entirety of its capacity. Before its roll out for clinical diagnosis, it is expected that there is ample space for the assay to be further optimized for even greater performance, based on the most recent advances in the CRISPR field. Many individual assay components can be optimized [15], such as the addition of RNase H, which was recently found to enhance the efficiency of RT-RPA leading to higher sensitivity of SHERLOCK [15]. To date, the assay can confidently detect CCHFV RNA at the 1 cp/µl concentration range (25 cp/reaction); however, a further upgraded formulation by incorporation of the newly identified optimizations should obtain an enhanced amplification activity and possibly reach a single-copy assay sensitivity (1 cp/reaction). On the level of the whole

formulation, the traditional two-step SHERLOCK protocol used here may be combined into a one-pot format integrating viral RNA release, pre-amplifications and CRISPR-Cas detection altogether, such as in the SHINE assay (Streamlined Highlighting of Infections to Navigate Epidemics) [15]. The single-tube format would allow multiple enzymatic reactions to run simultaneously. This enzymatic multitasking, together with the enhanced amplification activity of the overall formulation, should greatly cut down the run time needed to achieve a clear detection signal. The single-tube reaction can be further combined with fluorescent detection by a portable device, for example, a mobile phone [16,17], or colorimetric detection on a lateral flow strip, visualized directly by eyes [5,14]. These modifications are anticipated to streamline the assay into a simpler and faster test. Nevertheless, the degenerate sequence-based CRISPR set can server as the core of any CRISPR diagnostics targeting sequence diversity that can be shared among different assay formats.

In conclusion, the degenerate sequence-based CRISPR assay brings a functional alternative to traditional CRISPR diagnostics. While maintaining the performance achievable with a traditional constant-sequence CRISPR format, it offers additional advantages for detecting highly variable viral pathogens, and thus represents a useful new option in the toolbox of molecular diagnostic methods. Additionally, the new data generated by introduction of degenerate sequences into CRISPR for the first time may hopefully be of particular interest to the field and prompt future investigations towards new knowledge in CRISPR biology, including a clear understanding of the mismatch tolerance mechanisms. The resulting findings may lead to a greater fulfillment of the potential of CRISPR diagnostics.

## Supporting information

**S1 File. Supporting Figures A through G in a single PDF file.** This file contains references [63,64]. **Fig A.** DNA templates for in vitro transcription of the S segment from CCHFVs. **Fig B.** DNA templates for in vitro transcription of the S segment from CCHFV-related viruses. **Fig C.** Shannon entropy plot of 265 CCHFV isolates. **Fig D.** The degenerate sequence-based CRISPR diagnostic generates significant amplification signals rapidly at 1 cp/µl of CCHFV RNA from any clade/sub-clade. **Fig E.** Significant signal amplification with CCHFV but not closely related viruses. **Fig F.** Human genomic DNA contamination does not lead to false positive signals in the degenerate sequence-based CRISPR diagnostic for CCHFV. **Fig G.** Clade location of the IND and UAE isolates.
(PDF)

**S2 File. Phylogenetic tree of 265 CCHFV isolates (in a TIFF file).** This involved all the S segments that we found with a complete sequence described in GenBank. Phylogenetic analysis of these was performed by using the Maximum Likelihood method based on the General Time Reversible Model with Invariant sites using 1000 bootstrap replicates [64]. Branch lengths are measured in the number of substitution per site, with 1464 total nucleotide positions. The highest degree of sequence divergence was 20.152%. The calculation did not include MW464975, which is described in GenBank (accessed on 2021-11-03) as having a "complete" sequence, but contains unknown nucleotide types ("N"s in the listed sequence).
(TIFF)

**S3 File. Identity % matrix of 265 CCHFV isolates (in an excel file).** Multiple alignments of the CCHFV isolates in S2 File were conducted using Clustal Omega in Geneious Prime. Table lists nucleotide identity % between each pair of viral isolates. The highest degree of sequence divergence was 20.152%. The calculation did not include MW464975, which is described in GenBank (accessed on 2021-11-03) as having a "complete" sequence, but contains

unknown nucleotide types ("N"s in the listed sequence).
(CSV)

## Acknowledgments

We would like to thank Charles Lewis for his assistance for cultivating CCHFV for this work.

## Author Contributions

**Conceptualization:** Hongzhao Li, Bradley S. Pickering.

**Data curation:** Hongzhao Li, Bradley S. Pickering.

**Formal analysis:** Hongzhao Li, Dominic M. S. Kielich, Bradley S. Pickering.

**Funding acquisition:** James E. Strong, Bradley S. Pickering.

**Investigation:** Hongzhao Li, Alexander Bello, Greg Smith, Dominic M. S. Kielich, Bradley S. Pickering.

**Methodology:** Hongzhao Li, Alexander Bello, Greg Smith, Dominic M. S. Kielich, Bradley S. Pickering.

**Project administration:** James E. Strong, Bradley S. Pickering.

**Resources:** Alexander Bello, James E. Strong, Bradley S. Pickering.

**Supervision:** James E. Strong, Bradley S. Pickering.

**Validation:** Hongzhao Li, Dominic M. S. Kielich, Bradley S. Pickering.

**Visualization:** Hongzhao Li, Dominic M. S. Kielich, Bradley S. Pickering.

**Writing – original draft:** Hongzhao Li, Alexander Bello, Greg Smith, Dominic M. S. Kielich.

**Writing – review & editing:** Hongzhao Li, Alexander Bello, Greg Smith, Dominic M. S. Kielich, James E. Strong, Bradley S. Pickering.

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
