## [Decision Letter · Decision Letter 0]

23 Jul 2021

Dear Dr. Pickering,

Thank you very much for submitting your manuscript "Degenerate sequence-based CRISPR diagnostic to cover viral diversity" for consideration at PLOS Neglected Tropical Diseases. As with all papers reviewed by the journal, your manuscript was reviewed by members of the editorial board and by several independent reviewers. In light of the reviews (below this email), we would like to invite the resubmission of a significantly-revised version that takes into account the reviewers' comments. 

We cannot make any decision about publication until we have seen the revised manuscript and your response to the reviewers' comments. Your revised manuscript is also likely to be sent to reviewers for further evaluation.

Sincerely,

Brett M. Forshey

Associate Editor

Emma Wise

Deputy Editor

As you see below, while the reviewers clearly thought this CRISPR-based assay was an interesting approach to CCHFV diagnostics, they also raised a number of concerns about this manuscript, including data presentation (e.g. move Tables 1 and 2 to text) and the need to better support the idea that these assays exhibit extreme intolerance to sequence mismatches. Please respond to these concerns and determine what changes are needed in a revised manuscript.

Reviewer's Responses to Questions

**Key Review Criteria Required for Acceptance?**

**Methods**

-Are the objectives of the study clearly articulated with a clear testable hypothesis stated?

-Is the study design appropriate to address the stated objectives?

-Is the population clearly described and appropriate for the hypothesis being tested?

-Is the sample size sufficient to ensure adequate power to address the hypothesis being tested?

-Were correct statistical analysis used to support conclusions?

-Are there concerns about ethical or regulatory requirements being met?

Reviewer #1: (No Response)

Reviewer #2: The use of degenerate sequences for successful CRIPR-Cas13a detection of genomic variants of Crimean Congo Hemorrhagic Fever virus CCHFV is presented by Li H et al. The results were obtained using synthetic RNA targets.

Only one genome type of one representative of CCHFV was utilized for confirmation of the technique. The paper shows a potential technique to detect CCHFV which will include genomic variations previously reported. 

Degenerate sequence-based CRISPR diagnostic to cover viral diversity as a title would be too general, as the paper describes a specific example of diversity in the genomes of CCHFV.

The proposed objective to apply degenerate spacer region for the detection of other viral genomes was not provided, data regarding CCHFV shows a successful approach. The title may reflect the work specifically done for CCHFV detection.

Reviewer #3: Methods are clearly explained and well defined.

**Results**

-Does the analysis presented match the analysis plan?

-Are the results clearly and completely presented?

-Are the figures (Tables, Images) of sufficient quality for clarity?

Reviewer #1: (No Response)

Reviewer #2: The results could be summarized in two to three tables and figures, tables provided can be combined. Results can be compressed. The off-target detection of related viruses did not include other closer viral genomes, instead included very distantly (phylogenetically) viruses to prove specificity of the method. There is no nucleotide sequence homology among those viruses.

Figure 2 lists several gene bank accession numbers for the S region, but U04958.1 corresponds to the N region. Please clarify which accession numbers were for S and include most other distantly related CCHFV S sequences in the analysis.

Reviewer #3: authors have described well and presented. Table 1 and 2 are not needed it should be in text format. Figure 1 should be improved

**Conclusions**

-Are the conclusions supported by the data presented?

-Are the limitations of analysis clearly described?

-Do the authors discuss how these data can be helpful to advance our understanding of the topic under study?

-Is public health relevance addressed?

Reviewer #1: (No Response)

Reviewer #2: The main limitation found was that there are a large number of virus isolates with as low as 86% nucleotide identity (e.g. MH396666.1 from India) that should have been included in the BLAST comparison of Figure 2? what is the degree of nucleotide variation found amount all known CCHFV S gene regions?

Reviewer #3: How author can reduce time of detection using CRISPR-Cas13 system ?

Conclusions and their future perspective should be written clearly.

**Editorial and Data Presentation Modifications?**

Reviewer #1: (No Response)

Reviewer #2: Minor modifications include correction of few typos (line 310), and simplifying the sentences throughout the results and discussion sections. Also, Abstract narrative should not be repeated in the Introduction section.

**Summary and General Comments**

Reviewer #1: In this study, Hongzhao et al. proposed and aimed to test a design of CRISPR-based in vitro diagnostic assay based on degenerate gRNA. Even though it's an interesting idea, some major pieces of information were missing in the study which significantly compromised the quality and conclusions of the study. 

1) The authors claimed that CRISPR-based assays were highly intolerant to mismatches, and therefore a degenerate gRNA would be needed for viral detection. However, no experiments at all were performed to validate such a presumption, other than a few citations. Speaking from our own experience, there was a decent degree of mutation tolerance in these assays which allowed detection of pathogens even in the presence of natural variations. This presumptive hypothesis, which constituted the foundation of the current study, has to be carefully tested before introducing degenerate designs. 

2) There appeared to be a logic paradox between the mismatch-intolerant theory and the assay design. If a single mismatch between the gRNA and the target would abrogate the binding and hence the detection, the concentration in a degenerate gRNA design would need to be much higher than a single-sequence one in order to reach the same effective concentration. Say only to cover 4 degenerate NTs, a degenerate gRNA pool would consist 256 (4x4x4x4) difference sequences and only the one that match exactly to the target would work, which made the effective concentration as low as 1/256. In this study, the authors did not provide evidence proving that the degenerate design indeed contribute to the detection. 

3) The design of analytical assessments of an diagnostic assay needs to be improved. For instance, assay's LoD should not be evaluated based on the statistical difference between positive and negative samples. Rather, an assay cutoff should first be established and then technical repeats should be performed to determine the concentration at which at least 90% of the repeats were positive. 

4) No clinical specimens were tested at all. To develop a diagnostic assay, the sample type it could be used in should be defined and validated, due to the fact assay performance could vary greatly in various samples. Also, interference of human DNAs is another factor which should be considered and tested in both analytical, mock settings as well as clinical samples.

Reviewer #2: Overall, the paper proposes a potentially good molecular method to detect CCHFV. The paper may benefit from including more detailed comparisons of existing genomic variances. The list provided intends to include the most recent and diverse variants?

Reviewer #3: I have gone through and found manuscript is interesting for reader and scientific community. Before acceptance for publication, few minor revision needs to be addressed.

1. English and typo should be checked in entire ms.

2. Abstract should be improved by English for example “The same strategy may also be applicable to other types of CRIRPR diagnostics including those with Cas12a, Cas12b, Cas9 or Cas14…..”

3. RPA and CRISPR detection section table 1 and 2 should be written in text donot require in table form.

4. How author can reduce time of detection using CRISPR-Cas13 system ?

5. Conclusions and their future perspective should be written clearly.

PLOS authors have the option to publish the peer review history of their article (what does this mean?). If published, this will include your full peer review and any attached files.

Reviewer #1: No

Reviewer #2: No

Reviewer #3: No
---

## [Decision Letter · Decision Letter 1]

10 Feb 2022

Dear Dr. Pickering,

Thank you very much for submitting your manuscript "Degenerate sequence-based CRISPR diagnostic for Crimean–Congo hemorrhagic fever virus" for consideration at PLOS Neglected Tropical Diseases. As with all papers reviewed by the journal, your manuscript was reviewed by members of the editorial board and by several independent reviewers. The reviewers appreciated the attention to an important topic. Based on the reviews, we are likely to accept this manuscript for publication, providing that you modify the manuscript according to the review recommendations. 

Sincerely,

Brett M. Forshey

Associate Editor

Emma Wise

Deputy Editor

The reviewers had a couple of additional major comments on this revised draft:

- Reviewer 1 still has concerns about the nucleotide specificity of the gRNAs and their concentration and the implications for assay performance. I believe this comment is addressed in the Discussion to some extent already, but please review and determine if the concern can be addressed more comprehensively, since I would assume this would be a common question.

- Reviewer 4 mentions the need for a comparator assay. I do not think this require going back and running RT-PCR to compare, unless that is straightforward to do. However, I think it would be worthwhile figuring out if some sort of comparator can be provided. If comparator data was already generated during CRISPR assay design, consider including it, even if only in the Discussion. If that data is not already availabe, consider if there is some other way to introduce a comparison, as part of the Discussion.

- Reviewer 4 also had a minor comment about explaining how the copy number per ul was determined, which should be added to a revised version.

Reviewer's Responses to Questions

**Key Review Criteria Required for Acceptance?**

**Methods**

-Are the objectives of the study clearly articulated with a clear testable hypothesis stated?

-Is the study design appropriate to address the stated objectives?

-Is the population clearly described and appropriate for the hypothesis being tested?

-Is the sample size sufficient to ensure adequate power to address the hypothesis being tested?

-Were correct statistical analysis used to support conclusions?

-Are there concerns about ethical or regulatory requirements being met?

Reviewer #1: (No Response)

Reviewer #4: I have no concerns with the Methods section. The Methods are clearly stated in appropriate detail.

**Results**

-Does the analysis presented match the analysis plan?

-Are the results clearly and completely presented?

-Are the figures (Tables, Images) of sufficient quality for clarity?

Reviewer #1: (No Response)

Reviewer #4: - Please check over the results section on lines 381-391. How was the copy number per ul determined? Based on the sample input volume used in the reaction (line 183), 0.5 ul RNA was used in the reactions, but 1 copy/ul was reproducible detected (line 384). If 0.5 ul were used, the copy number in the reaction would be < 1.

- The manuscript would be greatly strengthened if the CCVHF dilutions/LODs were run with a comparator assay (ex. the Altona assay referenced in lines 531-532). Showing a LOD comparison would give the potential enduser a better idea about how your assay compares for more commonly used real-time RT-PCR.

**Conclusions**

-Are the conclusions supported by the data presented?

-Are the limitations of analysis clearly described?

-Do the authors discuss how these data can be helpful to advance our understanding of the topic under study?

-Is public health relevance addressed?

Reviewer #1: (No Response)

Reviewer #4: I have no concerns about the Conclusions. The limitations are clearly stated, and the overall study implications are not exaggerated. The public health relevance is addressed.

**Editorial and Data Presentation Modifications?**

Reviewer #1: (No Response)

Reviewer #4: - Table 3: add “virus” and the abbreviation for consistency. 

- Please check over the use of virus names and either spell them out or use the abbreviation. The use throughout the manuscript is inconsistent.

**Summary and General Comments**

Reviewer #1: As regular, single sequence CRISPR-based assays already showed a sufficient level of mismatch tolerance, the need for using degenerate sequences was not justified. In the revision, the authors acknowledged the above issue, but stated that the degenerate version enhanced the assay. However, lack of analytical and clinical evidences have been showed to support the statement and its significance. In fact, if the assay is highly dependent on the nucleotide specificity of the gRNAs, the degenerate assay would not work due to the very low gRNA concentration of each sequence variant.

Reviewer #4: The manuscript by Li and colleagues is an interesting study applying CRISPR to the detection of genetically diverse pathogens. This proof of concept study describes the incorporation of degenerate nucleotides into CRISPR-based diagnostic assays. The manuscript is well written, and stating this is a proof of concept study and highlighting limitations is greatly appreciated! The manuscript would be strengthened if some kind of a comparator assay were included in the assay LODs. This would provide the reader with an idea about how the sensitivity compares to real-time RT-PCR or something similar.

PLOS authors have the option to publish the peer review history of their article (what does this mean?). If published, this will include your full peer review and any attached files.

Reviewer #1: No

Reviewer #4: No

Figure Files:

Data Requirements:

Reproducibility:

References

---

## [Editor Report · Decision Letter 2]

27 Feb 2022

Dear Dr. Pickering,

We are pleased to inform you that your manuscript 'Degenerate sequence-based CRISPR diagnostic for Crimean–Congo hemorrhagic fever virus' has been provisionally accepted for publication in PLOS Neglected Tropical Diseases.

Best regards,

Brett M. Forshey

Associate Editor

Emma Wise

Deputy Editor

---

## [Editor Report · Acceptance letter]

7 Mar 2022

Dear Dr. Pickering,

We are delighted to inform you that your manuscript, "Degenerate sequence-based CRISPR diagnostic for Crimean–Congo hemorrhagic fever virus," has been formally accepted for publication in PLOS Neglected Tropical Diseases.

Best regards,

Shaden Kamhawi

co-Editor-in-Chief

Paul Brindley

co-Editor-in-Chief
